# Coupling Environmental Whole Mixture Toxicity Screening with Unbiased RNA-Seq Reveals Site-Specific Biological Responses in Zebrafish

**DOI:** 10.3390/toxics11030201

**Published:** 2023-02-21

**Authors:** Christian I. Rude, Lane G. Tidwell, Susan C. Tilton, Katrina M. Waters, Kim A. Anderson, Robyn L. Tanguay

**Affiliations:** 1Department of Environmental and Molecular Toxicology, Oregon State University, Corvallis, OR 97331, USA; 2Pacific Northwest National Laboratory, Biological Sciences Division, Richland, WA 99354, USA

**Keywords:** passive sampling, *Danio rerio*, polycyclic aromatic hydrocarbons, mixtures, developmental, transcriptomics

## Abstract

Passive sampling device (PSD) extracts paired with developmental toxicity assays in *Danio Rerio* (zebrafish) are excellent sensors for whole mixture toxicity associated with the bioavailable non-polar organics at environmental sites. We expand this concept by incorporating RNA-Seq in 48-h post fertilization zebrafish statically exposed to PSD extracts from two Portland Harbor Superfund Site locations: river mile 6.5W (RM 6.5W) and river mile 7W (RM 7W). RM 6.5W contained higher concentrations of polycyclic aromatic hydrocarbons (PAHs), but the diagnostic ratios of both extracts indicated similar PAH sourcing and composition. Developmental screens determined RM 6.5W to be more toxic with the most sensitive endpoint being a “wavy” notochord malformation. Differential gene expression from exposure to both extracts was largely parallel, although more pronounced for RM 6.5W. When compared to the gene expression associated with individual chemical exposures, PSD extracts produced some gene signatures parallel to PAHs but were more closely matched by oxygenated-PAHs. Additionally, differential expression, reminiscent of the wavy notochord phenotype, was not accounted for by either class of chemical, indicating the potential of other contaminants driving mixture toxicity. These techniques offer a compelling method for non-targeted hazard characterization of whole mixtures in an in vivo vertebrate system without requiring complete chemical characterization.

## 1. Introduction

The Portland Harbor Superfund Site (PHSS) was designated by the U.S. EPA in 2000 along approximately nine miles of the Willamette River flowing through Portland, Oregon due to the widespread contamination of sediments by chemicals including polychlorinated bisphenols (PCBs), dioxins, DDT degradation products, and polycyclic aromatic hydrocarbons (PAHs) [1]. Since then, the area has undergone significant remediation including dredging and capping at particularly contaminated sites [1]. Environmental monitoring and remediation efforts continue to this day and are expected to continue at least into the late 2020s [2].

The Oregon State University Superfund Research Program pioneered a technique utilizing low density polyethylene passive sampling devises (LDPE-PSDs) and developmental toxicity assays in *Danio rerio* (zebrafish) to measure toxicant abundance and bioactivity at numerous sites of interest within the PHSS [3,4]. Dechorionated developing zebrafish provide an excellent vertebrate biosensor of toxicity because they are easily reared, high throughput, and exquisitely sensitive to insult by bioactive compounds [5]. PSDs collect a time weighted average of the freely dissolved and therefore bioavailable fraction of contaminants at a site [6,7]. Extracts from PSDs yield whole environmental mixtures for chemical and toxicological characterization. The technique uses PSDs deployed in pairs. One PSD is spiked with performance compounds for more precise chemical measurement while the other remains unspiked to prevent interference in the bioassay. This technique was termed as Biological Response Indicators Devices Gauging Environmental Stressors (BRIDGES). Using it, Allan et al. found a significant correlation between the PAH concentrations and associated site toxicities in the PHSS [3].

Sediment within the PHSS contains high levels of PAHs, which in turn partition into the river water [1,8]. While not all PAHs appear to be toxic, many such as the infamous benzo(a)pyrene are classified as known or suspected human carcinogens, and others cause toxicity to the immune, cardiovascular, and neurological systems [9,10,11,12,13,14,15,16]. Canonically, many PAHs exert their effect through the aryl hydrocarbon receptor (AHR), a transcription factor that resides in the cytosol until a properly fitting ligand shifts its conformation [17]. It then breaks free from its chaperones and enters the nucleus where it canonically partners with AHR nuclear translocator (ARNT) to bind xenobiotic response elements within the genome. This induces the transcription of numerous genes involved in the xenobiotic response including p450s, wfkkn1n, foxq1a, and nrf2 [18,19,20,21,22].

PAHs are present in the environment as complex mixtures [23], however, traditional toxicity testing evaluates PAHs on an individual basis. BRIDGES is an ideal tool because it allows for toxicological evaluation of whole mixtures, even in the absence of complete chemical data. This study expands the concept of BRIDGES by incorporating RNA sequencing to interpret the molecular responses behind site specific toxicity in developing zebrafish. We set out to characterize the hazard of environmental mixtures, compare the molecular responses between PSD extract exposures, and determine how the molecular response to mixtures informs the drivers of toxicity.

## 2. Materials and Methods

### 2.1. Chemicals

Dimethylsulfoxide (DMSO), dichloromethane (DCM), hydrochloric acid (HCL), n-hexane, and isopropanol used for PSD precleaning and cleaning and chemical sample prep were Optima grade or better and purchased from Fisher Scientific. Performance reference compounds (PRCs) included perdeuterated flourene-D10, p,p’-DDE-D8, and benzo[b]fluoranthene-D10. Perdeuterated surrogate recovery standards included naphthalene-D8, acenaphthylene-D8, phenanthrene-D10, fluoranthene-D10, pyrene-D10, benzo[a]pyrene-D12, and benzo[ghi]perylene-D12. Samples were spiked with perylene-D12 before GCMS/MS analysis as an internal standard. The 33 PAHs included in the quantitative PAH method and their abbreviations are included in Appendix A. 

### 2.2. Sampling Methods

Lipid free PSDs were made from 2.7 × 100 cm LDPE tubing that was pre-cleaned with n-hexane and heat sealed at both ends, as described in Sower et al. [24]. PSDs intended for chemical analysis were spiked with deuterated reference compounds, while those intended for toxicity testing were not. At each sampling site, stainless steel cages containing five PSDs were deployed in matched pairs with one member of the pair intended for chemical analysis and the other intended for toxicity testing. Cages were secured ~3 m above the river floor with the anchored floatation setup described in Anderson et al. [25]. Each deployment lasted 30 days, at which point the PSDs were collected, sealed in amber glass jars, transported back to the lab in coolers, cleaned with HCL and isopropanol, and stored at −20 °C until extraction. 

The original sampling campaign has been described in detail by Allan et al. (2012) [3]. PSDs were deployed at nine locations: six within the Portland Harbor Superfund Site, two in the Columbia River, and one in the Willamette River above the Superfund Site. Sampling was conducted during the months of September and October 2009, and July–October in 2010 for a total of six different sampling periods. This study utilized extracts at river mile 7 West from September 2009 and river mile 6.5 West from July 2010 for chemical analysis and toxicological characterization. The extracts are referred to as RM 7W and RM 6.5W, respectively, throughout this study. 

The set of five PSDs from each cage were extracted together to yield one environmental extract per cage. The PSDs from cages intended for chemical analysis were spiked with perdeuterated PAH recovery standards before extractions to account for any losses. Extractions were carried out via two sequential dialyses in 40 mL of n-hexane per PSD. The first dialysis was 4 h long while the second was 2 h. The resulting 400 mL of dialysates were quantitatively concentrated to 1 mL before chemical analysis. The dialysates intended for toxicological testing were quantitatively solvent exchanged to dimethylsulfoxide (DMSO). 

### 2.3. Chemical Analysis

All chemical analyses were carried out on an Agilent 5975B gas chromatograph-mass spectrometer (GC-MS) equipped with a DB-5MS column (30 m × 0.25 mm, 0.25 μm).

PAH quantitation was accomplished in electron impact mode (70 EV) using selective ion monitoring. The GC injection port was held at 300 °C and the helium flow was 1.0 mL min^−1^ throughout. The oven temperature was as follows: initial 1 min hold at 70 °C, ramp rate of 10 °C min^−1^ to 300 °C, hold 4 min, ramp rate of 10 °C to 310 °C, hold 4 min. The MS quadrupole, source, and transfer line were held at 150 °C, 230 °C, and 280 °C, respectively. Each PAH was quantitated by the relative response to deuterated surrogate standards on a nine-point calibration curve with a minimum correlation coefficient of 0.98. Deuterated standard recoveries were between 44 and 109%. Lower recoveries occurred in the lower molecular weight deuterated standards due to volatile loss especially during the sample concentration. Target PAHs were recovery corrected based on the volatile loss of the deuterated recovery surrogates. The measured concentrations for each sample determined by the 33 PAH method are available in Appendix A.

The minimum detection limit for any of the PAH analytes was 10 pg/uL. Calibration controls were run every ten samples and were required to verify concentration within 15% of the true concentration to pass. All measurements reported in this study were preceded and followed by passing the calibration controls. Triplicate measurements from RM 7W were used to estimate the variance in both reported samples, as demonstrated by Matzke et al. [26].

The 1201 chemical screen was accomplished with the GC-MS running on full scan acquisition mode (mass range 5–50) and absolute EMV mode. The helium flow was 2.3 mL min^−1^ throughout. The oven temperature was as follows: initial 2 min hold at 70 °C, ramp rate 25 °C min^−1^ to 150 °C, ramp rate of 3 °C min^−1^ to 200 °C, ramp rate of 8 °C min^−1^ to 280 °C, hold 15 min, ramp rate of 40 °C min^−1^ to 310 °C, hold 3 min. The retention time was locked on chlorpyrifos at 19.23 min. The MS component temperatures were the same as the PAH method. 

The 1201 chemical screen included chemicals of concern from a broad array of chemical classes including, but not limited to, chlorinated bisphenols, parent and substituted PAHs, pharmaceuticals, phthalates, and synthetic musks. Detected chemicals were identified via their mass spectra using Agilent Deconvolution Reporting Software developed by Agilent utilizing the AMDIS spectral database [27]. The library of potential chemicals was limited to GC-MS compatible compounds with some degree of hydrophobicity, as would be required for sequestration to the LDPE-PSDs. 

Beyond the calibration controls, the quality control measures included laboratory preparation blanks, field and trip blanks corresponding to every deployment, laboratory cleanup blanks, and reagent blanks that accounted for approximately one third of all samples. All target compounds were below the LOD in the quality control samples.

### 2.4. Zebrafish Rearing, Exposure, and Morphological Assessment

Embryos used in the extract exposures were reared at the Sinnhuber Aquatic Research Laboratory from pathogen free tropical 5D strain zebrafish maintained according to the appropriate Institutional Animal Care and Use Committee (IACUC) approvals. Parent fish were housed in 100 gal tanks on a recirculating water system under 14–10 h light–dark cycle and fed twice a day with Gemma Micro 300. Water was maintained at 28 °C with a conductivity of 1000 μSiemens. Parent fish were group-spawned in the morning with the start of the light cycle. The resulting embryos were inspected for malformation, grouped by developmental stage, and enzymatically dechorionated at 4 h post fertilization (hpf). At 6 hpf, the embryos were transferred to a 100 µL exposure solution in 96-well plates, accessed again for viability, sealed, and incubated in the dark at 28 °C. 

For the morphologic toxicity assessments, developing zebrafish were statically exposed in E2 embryo medium (EM) with 1.00%, 0.20%, 0.04%, and 0.008% PSD extract concentrations. The 100% PSD extract in DMSO became a 100× PSD stock, while the 20×, 4×, and 0.8× stocks were made by serial dilution in DMSO. Exposure solutions were made by a 1:100 addition of PSD stock into EM, resulting in 1% DMSO solutions. The most concentrated exposure corresponded to ~1000 times the contaminate levels in the corresponding river water. Each concentration of PSD extract was tested in 40 fish in two separate plates. Each plate contained 20 embryos exposed at one of the four concentrations of PSD extracts, eight negative controls exposed to 1% DMSO, and eight positive controls exposed to trimethyltin at 5 μM. At 5 μM, trimethyltin elicited morphological malformation in 100% of embryos, but less than 20% mortality at 120 hpf. Plates failed quality control if mortality exceeded two fish in either of the controls. Failure of either the positive or negative control, or inconsistency between the two exposure plates, resulted in a retest. 

At 24 hpf, the developing zebrafish were assessed for mortality, delayed development, and notochord malformations. Then, at 120 hpf, the larvae were evaluated for mortality, abnormal touch response, yolk sac edema and pericardial edema, craniofacial abnormalities, and malformations of the body axis, brain, lower trunk, muscles, skin, and notochord. A wavy notochord malformation was also observed and included as part of the notochord endpoint. The dose response data for each exposure concentration is included in Appendix A.

### 2.5. RNA Isolation

The transcriptomic data generated from zebrafish larval exposures to PSD extracts were new to this study while that of the oxygenated PAH (OPAH) and parent PAH compounds have previously been published [21,28]. A summary table of the exposure scenarios and techniques to generate the RNA-Seq data is included in Appendix A. The PHSS extract and OPAH RNA samples were generated via the batch static, waterborne exposure of 20 embryos in 2 mL of solution in amber glass vials. For each PHSS sample, a ×75 PSD stock solution was diluted 1:100 in EM to yield a 0.75% PSD exposure solution. Embryos were exposed to phenanthraquinone (PHEQ) at 1.2 μM. Benz(a)anthracene quinone (BAAQ) and benzanthrone (BEZO) were used at exposure concentrations of 10 μM, which corresponded to the concentration producing a 100% effect at 120 hpf but no effect at 48 hpf. We also generated vehicle controls by batch exposing embryos to 1% DMSO. For each treatment, three groups of 20 fish were homogenized in RNAzol (Molecular Research Center, Inc.), using 0.5 mm zirconium oxide beads (Next Advance) in a bullet blender. Samples were stored at −80 °C until phenol guanidine extraction. The RNA concentration was determined with a Synergy MX microplate reader. The quality was determined with an Agilent Bioanalyzer 2100. We confirmed a RIN score above 9 in each sample before sending the total RNA samples to The University of Oregon Genomics Core, where RNA was poly-A selected via the Dynabead mRNA Purification Kit (Invitrogen), library prepped with the ScriptSec v2 Kit and ScriptSeq index primers, and 50 bp paired-end sequenced with an Illumina HISEQ 2000 sequencer.

This study also utilized the transcriptomic data measured from larval exposures to the parent PAHs retene (Ret), benzo(k)fluoranthene (BkF), and benzo(b)fluoranthene (BbF) originally generated and published by Shankar et al. [21]. We extracted the raw fastqs of these exposures from the GEO database, ascension number GSE171944. The exact generation of mRNA measured in these files is detailed in the aforementioned study, but briefly: dechorionated zebrafish were individually exposed to Ret, BkF, BbF, or 1% DMSO vehicle control in 96-well plates from 6 to 48 hpf. The concentrations used in the RNA generation exposures were equivalent to those that elicited 80% embryos exposed from 6 to 120 hpf. We refer to these concentrations as the EC80. At 48 hpf, fish were pooled into groups of eight, homogenized in RNAzole with 0.5 mm zirconium beads in a bullet blender, and purified with the Direct-zol Miniprep Kit including the optional in-column DNASE digestion step. After verifying the RNA purity and quality, the total RNA was sent to Oregon State University’s Center for Genome Research and Biocomputing for poly-A enrichment, library prep, and 100 bp multiplexed sequencing on an Ilumina Hiseq 3000.

### 2.6. Sequencing Data Pipeline

Beginning with raw fastqs, reads were accessed for quality using FASTQC [29]. Reads were not trimmed because the initial read trimming trials did not show improvement mapping to the genome. Fastqs were mapped to the GRZ11 genome using STAR Aligner, and reads were tallied with HTSEQ [30,31]. Further analysis was accomplished in R [32]. Unless otherwise stated, graphs were made using ggplot2. We used multidimensional scaling of the raw reads to identify outliers within the treatment groups, which resulted in the exclusion of a single treatment from the treatment groups RM 6.5W, BbF, and one vehicle control. Each of these groups was reduced to n = 2, while the remaining treatment groups analyzed in this study had n = 3 or n = 4 samples. Differential expression was determined with DESEQ2 using all treatment groups to estimate common dispersion [33]. The resulting log_2_foldchange (Log_2_FCs) and Benjamin–Hochber adjusted *p*-values (p_adj_) were used to determine the differentially expressed genes (DEGs) for further analysis. All DEGs with determined during analysis are included with thier log_2_FC and p_adj_ values in Appendix A.

### 2.7. RNA-Seq Analysis

All heatmaps were made with pheatmap in R [32]. The gene expression heatmaps showed the log_2_FC of every gene defined as a DEG (|log_2_FC| ≥ 1, p_adj_ ≤ 0.05) in at least one of the displayed treatments. In the direct PHSS extract comparison, rows were clustered by Euclidian distance. In the comparison with single chemical treatments, the rows were clustered via k-means clustering with k = 8 and 500 starts. Gene Ontology analysis was performed using g:profiler2 in R. We used the Gene Ontology (GO) databases for Biological Processes (BP), Cellular Components (CC), and Molecular Function (MF) [34,35]. Databases were filtered to GO terms that included 15–400 genes to remove overly broad or narrow terms. We broadened our analysis by reducing the |log_2_FC| cutoff for DEGs to 0.5. The GO terms were tested for overrepresentation by a hypergeometric test and corrected for multiple tests using the built in “g_SCS” functionality. Network analysis of the GO terms was performed using the Cytoscape app Enrichment Map clustering GO terms by shared gene sets [36,37]. To generate the GO term heatmap, significantly enriched GO terms were ranked by the enrichment ratio within each treatment and database. The top 60 GO terms from each treatment were then clustered hierarchically by Jaccard distance. GO terms with Jaccard distances < 0.45 were combined, with the combined term, keeping the name of the smallest GO term. We utilized the reduced set of GO terms to produce the heatmap. 

## 3. Results and Discussion

### 3.1. Chemical Characterization of PSD Extracts: Parallel PAH Contamination, Divergent Qualitative Screen

Parallel PSD extracts allowed for exquisite PAH characterization using deuterated PRS and surrogate standards in the analytical extracts without fear of interference in the developmental toxicity assays. Undiluted extracts contained the chemicals concentrated in five HDPE strips over 30 days at each of the sampling locations. We report the nominal PAH concentrations in the 1% extract exposures. These PAH concentrations were approximately 1000 times higher than the concentration in the PHSS and were intended to identify differences in toxicity between mixtures, rather than mimic the environmental exposures. The sum PAH concentrations (Figure 1A) in each sample were 190 ± 13 and 310 ± 20 μM (±95% CIs) in RM 7W and RM 6.5W, respectively. The concentrations of each PAH above the limit of quantification are displayed in Figure 1B. Despite the 60% difference in magnitude, the PAH ratios between the two samples were nearly identical for FLA/PYR, RET/CHR, and PHE/ANT (Appendix A). Given the similar ratios, it is likely that the PAH contamination sources are similar, if not the same, at the two sites. Thus, even though the quantitative chemical analysis was limited to 33 PAHs, the composition of all PAHs in the two mixtures are likely very similar.

The 1201 chemical screen detected three chemicals in both extracts and five chemicals unique to only RM 7W (Table 1). Two DDT degradation products o,p’DDD and p,p’-DDE, along with hexachlorobenzene were detected in both samples. PCB65, PCB118, the synthetic musk tonalide, and two OPAHs benzofluorenone and BEZO. DDT degradants and hexachlorobenzene are both known PHSS contaminants, likely due to their use as pesticides [1]. More surprising is that despite having similar PAH profiles and higher concentrations of PAHs in RM 6.5W, there were more chemicals detected in RM 7W. While the PAH contaminant sources are similar, this is not necessarily true for other contaminants that may have contributed to the extract toxicity.

### 3.2. Developmental Toxicity: RM 6.5W Was More Toxic Than RM 7W

Developmental toxicity assays demonstrated that RM 6.5W was more toxic than RM 7W and identified notochord malformations as the most sensitive developmental endpoint. Dechorionated zebrafish embryos were statically exposed to the vehicle control or to four dilutions of each PHSS extract and observed for negative outcomes at 24 and 120 hpf as a binary hit or no-hit response. As evident in Figure 2A, neither extract exerted developmental toxicity significantly different from the controls by 120 hpf at the two lowest concentrations, and both extracts exerted 100% incidence of malformation (“any effect except mortality”) or mortality at the 1% exposure. In contrast, exposure to RM 6.5W at the 0.2% dilution caused malformations in 67.5% of the tested embryos, significantly more than RM 7W, which caused malformations in 26.5% of fish (Fisher’s exact, *p* < 0.05). This data indicate that the developing zebrafish were more sensitive to RM 6.5W than RM 7W, and therefore, the nonpolar bioavailable fraction of contaminants at RM 6.5W was more toxic than the RM 7W site for the PSD deployment dates.

The increase in any effect except mortality from RM 6.5W compared to RM 7W and the controls was driven primarily by the wavy notochord malformation (Figure 2B), an unusual occurrence in developing zebrafish and not characteristic of PAH or AHR ligand exposure. Representative images for these phenotypes are included in the results in the study by Allan et al. [3]. In a developmental screen of 123 PAH and PAH-derivates, this malformation was not associated with any parent or alkyl PAHs and was only associated with four OPAHs, only one of which was an AHR ligand [38]. While two OPAHs were detected in RM 7W, neither elicited the wavy notochord response. The additional contaminants from RM 7W did not increase the toxicity of this mixture compared to RM 6.5W. Instead, the increased toxicity of RM 6.5W was consistent with its greater PAH concentration. Since PAHs alone did not explain the phenotypes observed in the developing embryos, it is likely that other toxicants in the mixture mirrored, to some degree, the PAH concentrations to drive toxicity.

### 3.3. RNA-Seq: Direct Comparison of PHSS Extract Responses

#### 3.3.1. RM 6.5W Elicits a More Robust Gene Expression Response than RM 7W

RM 6.5W elicited a greater transcriptional response than RM 7W in embryos exposed from 6 to 48 hpf. We conducted RNA-Seq on three independent pools of 20 fish for RM 6.5W and RM 7W exposed at the 0.75% extract and an equal number of vehicle controls exposed to 1% DMSO. This extract dilution was chosen because it caused a high percentage of effect in larvae at 120 hpf without a significant effect at 48 hpf in both the extract treatments. These were exposed at equal extract dilutions so that the gene expression would reflect the relative bioactivity of each mixture. 

Exposure to RM 7W resulted in 506 DEGs (|logFC| > 1 and p_adj_ < 0.05) while exposure to RM 6.5W resulted in 963 DEGs (Figure 3A). The directionality of the gene expression changes between the two samples was well-conserved. We compared the gene expression responses associated with exposure to each sample by plotting the differential expression in a heatmap, as shown in Figure 3C. There were almost no significant gene expression changes with different directionality between the two samples. Of the 1003 DEGs significant in either of the treatments, only 53 had different directional changes, and of these, only two were significant in both treatments. This was similar to the number of false positives we expected given p_adj_ = 0.05. The deeper colors in the RM 6.5W column illustrate the larger transcriptional responses to RM 6.5W.

To compare the magnitudes of gene expression, we plotted the log_2_FC of genes with differential expression in either of the samples in Figure 3B. The expression patterns associated with the two samples were highly correlated. Fitting a linear model through the origin yielded a slope of 0.59 ± 0.02 (95% CI) with an r^2^ = 0.74. This clearly indicated stronger gene expression changes resulting from the RM 6.5W exposure rather than the RM 7W exposure. The strong correlation in gene expression between the two mixtures was evidence that they exerted toxicity in a similar fashion. It is likely that the greater toxicity of RM 6.5W compared to RM 7W was caused by the same or similar components at higher concentrations affecting similar toxic pathways with more intensity. 

The greater gene expression changes associated with RM 6.5W and the correlations in differential expression between the two exposures were consistent with those observed in the concentration–response evaluations. Namely, at the same dilution factors, the more toxic mixture evoked a larger transcriptional response. Used in this simple top–down approach, transcriptional responses in developing zebrafish exposed to PSD extracts clearly identified the more toxic mixture without having to exhaustively measure the mixture components. Up to this point, PAH concentrations have been associated with mixture toxicity, but have not been proven to drive it. Next, we used the DEGs to identify the potential negative outcomes to the developing embryos caused by these mixtures and their consistency with known PAH toxicity.

#### 3.3.2. Distinct Themes Underly the Transcriptional Response to PHSS Extract Exposures

We used a Gene Ontology network approach to identify concerted biological themes within the transcriptomic response to the mixtures. In this analysis, we decreased the log_2_FC cutoff for DEGs to |log_2_FC| < 0.5 and split the DEGs into three groups: DEGs specific to RM 7W treatment (RM 7W set), DEGs shared by both treatments (overlap set), and DEGs specific to RM 6.5W treatment (RM 6.5W set). We tested for overrepresentation in these DEG sets using terms from the Gene Ontology Database (GO) and visualized the results using the Cytoscape app Enrichment Map. The resulting network is displayed in Figure 4. Each node is a GO term colored by gene set, and edges denote the relative quantity of genes shared between them. The nodes were arranged into distinct clusters. We circled and numbered clusters with more than three GO terms and reviewed the terms to determine the themes included in the table in Figure 4. The full network file is available in the Appendix A. 

Surprisingly, the RM 7W set yielded no significantly enriched GO terms despite including 190 DEGs. The simplest explanation of this is that the DEGs unique to the RM 7W exposure did not represent any concerted pathway. Conversely, the overlap set resulted in 40 enriched GO terms. This vast disparity indicated that toxic processes caused by both PHSS extracts were significantly more important to RM 7W toxicity than any RM 7W-specific effect. Therefore, it did not appear that the additional contaminants detected in RM 7W affected its toxicity.

Another major trend was that the GO terms did not tend to be enriched in more than one DEG set. With the exception of a single GO term in cluster 7, every term was either from the overlap set or RM 6.5W set, but never both. This is likely because, by definition, the two DEG sets did not overlap. For a GO term to be enriched in both sets, it had to do so with entirely separate genes from both sets of DEGs. 

The network distinguished between transcriptional responses unique to RM 6.5W exposure and those present in both PHSS extract exposures. GO terms diverged into distinct transcriptional clusters based on the shared genes in their terms. We divided clusters by which the DEG set tended to dominate among their GO terms. In this way clusters 3, 5, and 6 represent transcriptional themes elicited by exposure to both PHSS extracts, and clusters 1, 2, 4, and 8 are themes that emerged specifically out of RM 6.5W toxicity. These RM 6.5W emerging clusters could arise as downstream steps resulting from greater perturbation of earlier shared clusters, or they could represent completely distinct toxic processes to RM 6.5W. While not a certainty, the high correlation in transcriptional changes, even among genes that only met the DEG cutoff in RM 6.5W, makes the former explanation more likely. In order to decisively determine between them, we would need RNA measurements from embryos exposed to a phenotypically anchored concentration such as an EC80.

#### 3.3.3. Xenobiotic Metabolism Glutathione Processes Are among the Expected GO Network Clusters

Clusters 5 and 7 contained GO terms and gene expression commonly associated with PAH exposure. Cluster 5 was bimodally distributed with terms relating to oxidant detoxification and xenobiotic metabolism. These two modes were connected by four GO terms, two from the RM 6.5W set and two from the overlap set, all dominated by genes for proteins containing hemes. Cluster 5 had the largest mean |log_2_FC| of any of the clusters, driven largely by classic AHR-responding genes such as cyp1a1, cyp1b1, ahrra, and foxq1a. The differential gene expression for these and other genes we call out throughout the study are included in Appendix A. RM 6.5W and RM 7W both contained PAHs, many of which were AHR ligands that elicited these responses in cell cultures and whole animal models including developing zebrafish [19,21,39]. Additionally, AHR is known to increase the transcription of nrf2, also upregulated in this cluster, which in turn regulates the response of many genes featured in the oxidant response GO terms [22].

Cluster 7, termed hydrolase and glutathione metabolism, also contained terms from both the overlap set and RM 6.5W set. The three terms in this cluster directly related to glutathione metabolism were enriched in the overlap set, while the four GO terms enriched in the RM 6.5W set were lyase related terms. The two groups were bridged by the terms “one-carbon metabolic process” and “cellular modified amino acid process”, which each contained a few genes encoding lyases. The lyase-related GO terms did not provide easily interpretable insights because they were a broad class of enzymes and did not interrelate beyond having a similar catalytic mechanism. Glutathione related GO terms were similar to cluster 5. Glutathione is an important detoxifying agent [40]. As expected, all of the genes in this GO term, except for one, had increased transcription in response to both extract exposures.

#### 3.3.4. Visual System Development and Muscle Fiber Related Genes Are Disrupted

Clusters 5 and 7 might be viewed as part of a reversible response by the developing zebrafish to the toxicants in the mixture. In evaluating the mixture toxicity, it would be more helpful to focus on clusters that are more likely to represent irreversible adverse outcomes. Clusters 3 and 8 likely correspond to irreversible toxic outcomes. 

Cluster 3, visual perception and eye development, indicated significant visual system impairment. It was made entirely of GO terms from the overlap set, meaning that it was active in both PHSS extract exposures. Of the 60 genes in this cluster, 53 had decreased expression. Thirty-four were crystallin genes, which, when transcribed, form the proteins that make the lens of the eye [41]. Additionally, opsin 1, rhodopsin, and two other genes involved in photo transduction rom1b and arr3a had log_2_FC levels between −6.6 and −2.2 from RM 6.5W treatment, and −2.2 and −0.86 from RM 7W treatment, respectively. The widely decreased expression in crystallin genes and photo transduction genes indicated impaired eye development. This effect might have been mediated by Cyp1b1 expression induced by AHR. Cyp1B1 is naturally expressed in the eyes of zebrafish, mice, and humans, where it is known to play a role in development likely through the metabolism of a yet undetermined endogenous signaling molecule [42,43,44,45]. Despite this, the effects on the visual system are not classically implicated in AHR-mediated toxicity. While it is possible that the mixture of PAHs might have induced these effects through AHR, other constituents of the mixture could have acted through alternative means. 

Cluster 8, themed muscle fiber constituents, also likely represented an irreversible adverse outcome and was interesting in light of the wavy notochord malformation induced so strongly by RM 6.5W exposure. It was the only cluster in the network that consisted mostly of GO terms related to Cellular Components, with a total of 25 genes enriched in six muscle cell constituent related terms. Cluster 8 also includes the GO terms “actin cytoskeleton” and “muscle cell development”, which are broader in scope. Differential expression trended negatively, much like the visual perception cluster. For RM 6.5W exposure, the average log_2_FC of all DEGs in the muscle fiber cluster was −0.49 and 50 out of 60 DEGs had negatively impacted expression. The DEG with the largest fold change in this cluster was myl2b, an ortholog of human MLC2. It encodes a regulatory myosin light chain, which plays a role in potentiating muscle contraction and is essential for normal heart development in both zebrafish and mice [46,47]. Zebrafish with truncated mlc-2 or morpholino knocked-down mlc2 do not survive past seven days due to a lack of myofibril genesis [46]. With this in mind, the 5-fold reduction in mlc2 observed in embryos exposed to RM 6.5W very likely led to significant cardiac impairment. This cluster is particularly interesting in light of the wavy notochord phenotype caused by RM 6.5W exposure. Widespread disruption of structural components within these GO terms may have indicated diminished structural integrity within the embryos, resulting in a wavy notochord. PAHs are known cardiotoxins, with higher molecular weight PAHs typically acting through AHR signaling while lower molecular PAH toxicity is often independent of AHR; however, the widespread disruption of muscle fiber constituents has no precedence among PAHs [11,48,49]. The wavy notochord has previously been associated with a few OPAHS, two of which, 2-ethylanthraquinone and 3-hydroxyflouranthene, have parent PAHs measured in the PHSS extracts [38]. These data suggest that there are some PAH metabolites that can cause the phenotype. Exposure to a high concentration of a complex mixture of PAHs such as that of the PHSS extracts increases the likelihood that at least some of the metabolites could induce this phenotype. Conversely, exposure to a mixture of the ten most abundant PAHs in the PHSS extracts was not associated with this phenotype in previous studies [50]. The phenotype may also be caused by contaminants that went undetected by the 1201 chemical screen. In a screen of 1006 Phase 1 and Phase 2 Toxcast chemicals, the wavy notochord was only noted in exposure to 16 compounds, seven of which contained thiocarbamate functional groups [51]. In an earlier study, 0.8 μM exposure of metam sodium, a dithiocarbamate, resulted in the wavy notochord, and caused abnormal muscle physiology around the spinal cord [52]. A follow-up study demonstrated the wavy notochord in developmental zebrafish exposures in all nine dithiocarbamates tested [53]. This suggests that the gene expression changes in the muscle constituent cluster and wavy notochord could be driven by dithiocarbamate containing compounds. We recommend future chemical analysis conducted within the PHSS include targeted analysis for this class of compounds.

### 3.4. RNA-Seq: Comparison of Mixtures to Individual PAH and OPAH Exposure Responses

We next interpreted the gene expression of PHSS extract exposures through comparison to that of individual PAHs and OPAH. Ret, BkF, and BbF were selected because they are bioactive PAHs present in the mixtures, and OPAHs were included because they were detected in the RM 7W extract and are known to form during PAH degradation in the environment [54]. 

#### 3.4.1. Principal Analysis Indicates AHR as Second Strongest Determinants of Variation

PCA analysis distinguished the exposures by chemical class and AHR activation status, and showed that the PHSS extract exposures were more similar to OPAH exposures than to PAH exposures (Figure 5A). We defined DEGS has having |log_2_FC| > 1 and p_adj_ < 0.05 and used a PCA analysis to assess the similarity of treatments. Along PC1, the PAHs grouped together, followed by OPAHs, then RM 7W, and finally RM 6.5W. The OPAHS were more similar to RM 7W than to the PAHs, and RM 7W was much more similar to the OPAH exposures than to RM 6.5W. It is tempting to conclude that PC1 resolved by chemical class, with RM 7W grouping with OPAHs, but a parallel explanation was that PC1 divided by exposure toxicity. For instance, PAHs were exposed at the EC80 concentration, OPAHs at the EC100, and the PHSS exposures likely above the EC100. To differentiate between these two explanations, exposure concentrations would need to be normalized to the same EC before measuring gene expression. PC2 resolved the strong AHR activating exposures from the weak AHR activating exposures. PHEQ and BEZO are weak AHR transcription activators, as demonstrated by the cyp1a log_2_FC < 1.5, resulting from both exposures. In contrast, all other exposures in this study had a cyp1a log_2_FCs greater than 5. This trend was echoed for other AHR reporters such as cyp1b1 and ahrra. In the same way, BEZO and PHEQ were much lower than the rest of the exposures along PC2. PC1 accounted for 53% of the variance between samples while PC2 accounted for 20 % of the variance in gene expression. The ability of PC2 to resolve strong from weak AHR activators indicated that AHR contributed to a significant portion of the gene expression response, but given the much greater strength of PC1 and the much larger difference between the two PHSS extracts in PC1, there were clearly other factors more responsible for driving the differences in transcription and toxicity among the PHSS extracts.

#### 3.4.2. Differential Expression Heatmap Contains AHR Transcription and OPAH Related Clusters Yet PHSS Extract Exposure Remains Distinct

Figure 5B is a heatmap for any gene that was differentially expressed in at least one treatment. The exposures were clustered hierarchically according to their differential expression, and by k-means. Again, RM 7W clustered with the OPAHs, the PAHs clustered together, and RM 6.5W had the most unique gene expression. 

The differential expression from PHSS extract exposures followed the PAHs most closely in K2 and K3, but was largely distinct from these throughout the rest of the heatmap. K2 and K3 included many genes involved in xenobiotic metabolism and detoxification such as glutathione transferase ugt5a, P450s cyp1a1, cyp1a2, and cyp1a3, and other classic AHR response genes such as wfikkn1, ahrra, ahrrb, and the transcription factor foxq1a. In the remaining K groups, excluding K6, the average differential expression associated with PHSS exposures was higher in magnitude than that associated with individual PAH exposures. Similar to the relationship between RM 6.5W and RM 7W, the extra DEGs in the PHSS exposures might have resulted from differences in dosing. The zebrafish embryos were exposed to PAHs at the corresponding EC80 concentration, while the exposures to the PHSS extracts were likely at or above the EC100. Synergistic mixture effects among PAHs might also explain the differing expression. For instance, zebrafish co-exposed to the PAHs benzo(a)pyrene and fluoranthene, the former an AHR ligand, the latter a P450 inhibitor, experience a high level of cardiotoxicity not otherwise observed in exposure to either PAH alone [55]. We also cannot rule out other compounds within the mixture such as p-DDE and o-DDE, or compounds without good matches in the mass spectrum database as causative agents of these changes. Although DEGs induced by individual PAHs are present, the majority of transcriptional responses associated with exposure to the PHSS extracts was not accounted for by individual PAH exposure or single ligand–AHR activation. 

The DEGs of both PHSS extracts were more closely related to DEGs from the OPAH exposures. K1 and K4–K8 displayed similar gene expression changes among the OPAH and PHSS extracts. In each of these groups, the DEGs were greater in number and magnitude for RM 6.5W exposure, followed by RM 7W, and finally the OPAHs. Given that OPAHs were only detected in RM 7W, it seemed peculiar that the OPAHs mirrored the PHSS extract expression better than the PAHs. It is unlikely that this occurred due to erroneous RNA-Seq results because they largely agree with other existing gene expression results for these compounds (Appendix A). Differential expression measured with qRT-PCR of cyp1a, cyp1b1, and akr1c1 in hepg2 cells exposed to BAAQ mirrored our RNA-Seq results [56]. Furthermore, a previous study by our group utilized qRT-PCR to examine the expression of common xenobiotic response genes with exposure to OPAH. A comparison to those results showed no disagreement among the DEGs determined in both techniques [57]. The apparent conundrum might arise from limitations in the non-exhaustive 1201 chemical screen. It is possible that some OPAHs were present in RM 6.5W but were not part of the screen. Likewise, some other undetected chemical or chemicals could have induced the OPAH-like gene expression. Alternatively, as suggested earlier, parent PAHs may be metabolized to OPAHs in sufficient quantities within the embryos to elicit similar effects to OPAH exposure.

#### 3.4.3. There Is Significant Overlap between Individual Constituent and Mixture GO Terms but Muscle Fiber Related Genes Are without a Match

Finally, we hypothesized that GO term enrichment analysis could identify shared processes between the PHSS exposures and individual chemical processes. The analysis showed that GO terms resulting from both PHSS exposures could largely be found in individual OPAH and PAH exposures, while some RM 6.5W specific terms were still unaccounted for. Figure 6 summarizes the enrichment analysis results for the top 10 GO terms for each treatment excluding redundant terms. 

There were 24 unique GO terms enriched in the DEGs from at least one of the PHSS extracts, 18 of which were also present in at least one PAH or OPAH exposure. With the exception of one GO term, all terms shared between the two PHSS extracts were also elicited by PAH or OPAH exposure. The most shared GO terms were “response to oxidative stress”, “cellular response to xenobiotic stimulus”, and “heme binding,” all of which were terms from Cluster 5 in the network analysis. This indicates that many shared processes in the toxicity of the two PHSS extracts can be accounted for by shared or similar toxicants within the mixtures.

There were three visual perception related GO terms resulting from PHSS exposures, each of which was also associated with BEZO exposure. BEZO exposure resulted in 34 DEGs also present in the visual perception terms of the PHSS exposures. The linear model relating log_2_FC of these DEGs between BEZO and RM 6.5W was significant (*p* ≤ 0.05) with R^2^ = 0.9 and largely echoed by a model relating these DEGs between BEZO and RM 7W. Without filtering for significant DEGs in the BEZO exposure, there was still a significant correlation in the differential expression between the BEZO and the PHSS samples (*p* ≤ 0.05). BEZO is a weak AHR transcriptional activator, and cyp1b1 is not a DEG associated with BEZO exposure, however, BEZO was able to induce a remarkably similar toxicity to the visual system. While not confirming a mechanism, this indicates that disruption to the visual system associated with PHSS exposure may be Cyp1B1 independent.

While many GO terms enriched in DEGs from PHSS extract exposures were represented in the PAH or OPAH exposures, a few GO terms, most notably “actin cytoskeleton”, “troponin complex” and “sarcomere” were unique to RM 6.5W exposure. These terms harkened to the muscle cell constituent cluster from the network analysis. Their lack of enrichment in the DEGs from the other exposures increased the likelihood that this particular toxicity arose from non-AHR ligand constituents of the mixture. 

## 4. Conclusions

The trifecta of RNA-Seq, chemical analysis, and toxicity screening using PSD extracts and developing zebrafish proved to be a powerful approach to characterize hazards in whole environmental mixtures. Transcriptional responses provided a non-targeted method to identify perturbed biological processes underlying the gross malformations resulting from exposure to PHSS extracts. High correlation between differentially expressed genes in RM 7W and RM 6.5W indicated that similar xenobiotics affecting similar pathways drove the toxicity of both mixtures. Alone, the analytical chemistry and embryonic morphology assays detected adverse outcomes suggestive of PAH bioactivity, but the added dimension of transcriptomics uncovered not only PAH transcriptional signatures, but also perturbation of the visual and musculature systems uncharacteristic of canonical AHR ligand toxicity. Here, the differential gene expression of both PHSS mixtures was more similar to OPAHs than to PAHs, but also identified toxicities suggesting that mixture effects of other contaminants might be at play. Despite the suggestive gene expression data, true identification of the causal toxicants would likely require effects directed analysis, which was beyond the scope of this study.

The interpretation of our data was made more difficult because the exposures from which we measured the transcriptional responses were set at equal extract dilutions rather than to anchored to a phenotype. This resulted in the more toxic mixture exposure eliciting a larger transcriptional response, but came with an increased difficulty in specifying which transcriptional differences were truly unique between the two samples, and which differences were more likely to be due to the RM 6.5W exposure being “further up” the concentration–response curve. For the same reason, it also somewhat complicated our comparison to the differential expression resulting from individual compounds. To provide more certainty in discerning transcriptional differences, future studies would be better served by anchoring the mixture exposures to concentrations causing a predetermined level of phenotypic response such as the EC80. In typical circumstances where little is known about the mixture compositions, RNA-Seq could be used as a sort of fingerprint for the worst actors in the mix of chemicals. As more relevant comparative transcriptomic datasets between environmental mixtures and chemicals standards become available, such fingerprinting will become increasingly informative.

## Figures and Tables

**Figure 1 toxics-11-00201-f001:**
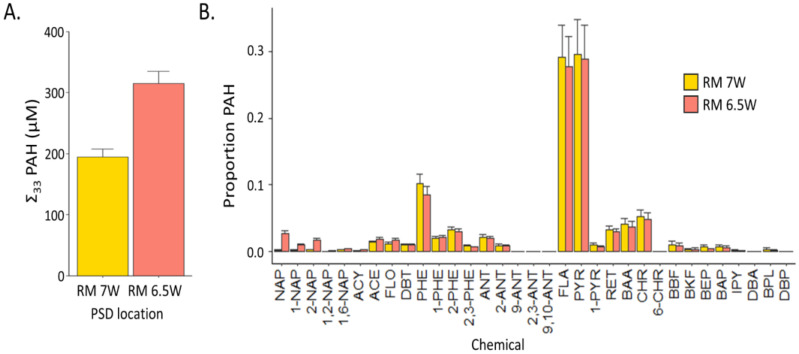
The results of the chemical analysis from the extracts of HDPE-PSDs deployed at RM 7W and RM 6.5 W during September 2009 and July 2010, respectively. The concentrations are reported at the expected levels in 1% extract exposures. (**A**) The sum uM of each PSD extract for the 33 PAHs included in the quantitative PAH method. (**B**) The proportion of each individual PAH measured in the 33 PAH method.

**Figure 2 toxics-11-00201-f002:**
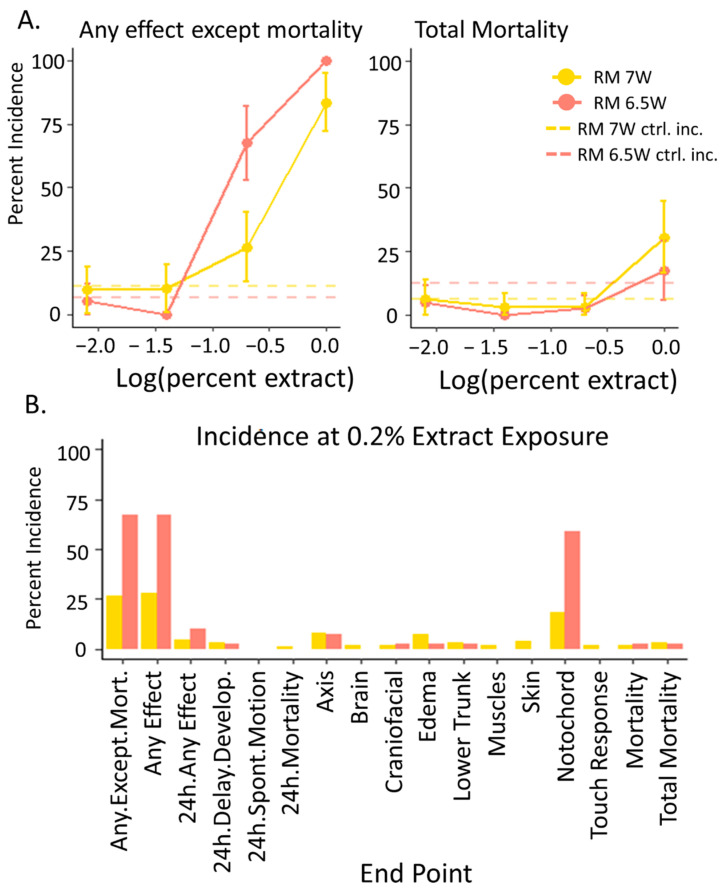
Concentration response data for the embryos exposed to the PSD extracts from RM 7W (September 2009) and RM 6.5W (July 2010) in the PHSS determined by the response of 40 developing zebrafish to each dose. (**A**) The percent incidence in the endpoints “any effect except mortality” and “mortality” in zebrafish by 120 hpf. Error bars indicate the 95% confidence intervals calculated utilizing the binomial distribution with n = 40. The hashed lines indicated the response levels in the vehicle controls associated with each mixture. (**B**) The observed percent incidence of every measured endpoint determined at the 0.2% extract exposure.

**Figure 3 toxics-11-00201-f003:**
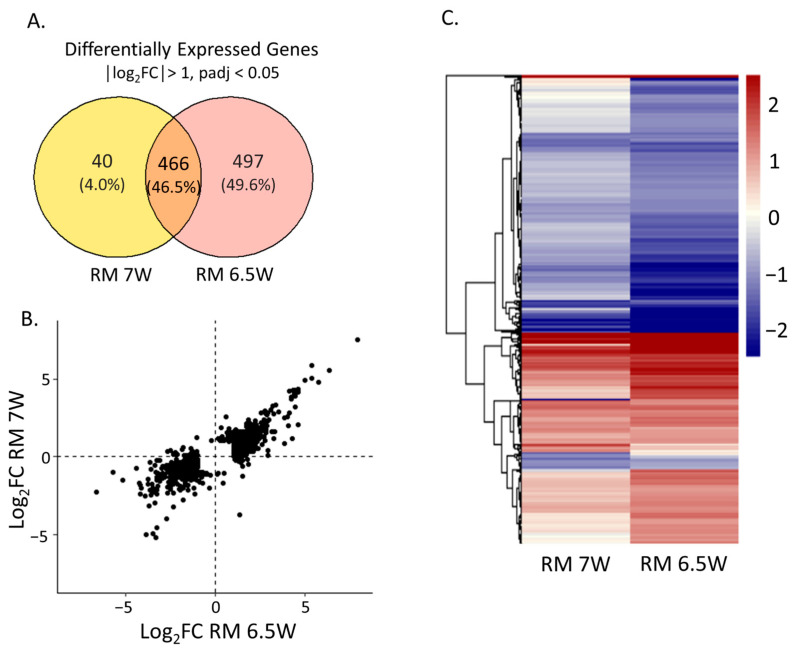
Differential expression in the embryos at 48 hpf after static exposure to the 0.75% extracts in embryo media from 6–48 hpf compared to the DMSO exposed control embryos. (**A**) The numbers of unique and shared differentially expressed genes (DEGs) meeting the threshold of │log_2_FC│ > 1, p_adj_ > 0.05 for each sample. (**B**) Log_2_FC of gene expression comparing the exposure conditions for any gene differentially expressed under at least one of the conditions. (**C**) Heatmap displaying the Log_2_FC of each sample, with genes clustered hierarchically by Euclidian distance.

**Figure 4 toxics-11-00201-f004:**
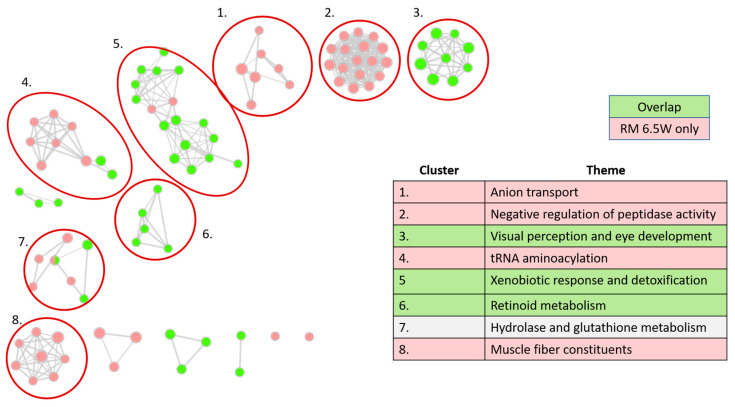
Gene Ontology (GO) network analysis. Each node is a significant GO term from one of three GO: Biological Processes (GO:BP), Cellular Components (GO:CC), or Molecular Functions (GO:MF). Ontologies were truncated to GO terms containing between 15 and 450 genes before computing significant adjuster enrichments on g.profiler2. Edges indicate the proportion of shared genes between terms, with an overlap coefficient (C_overlap_ = n_intersect_/n_genes in smaller term_) of at least 0.4. Node color indicates gene sets. Gene sets included DEGs meeting │log_2_FC│ > 0.5 and p_adj_ > 0.05, from one of three categories: DEGs specific to RM7 W, DEGs shared between the two treatments, and DEGs specific to RM 6.5W. There were no significant GO terms enriched in the gene set specific to RM 7W. The table indicates the manually determined themes for each cluster.

**Figure 5 toxics-11-00201-f005:**
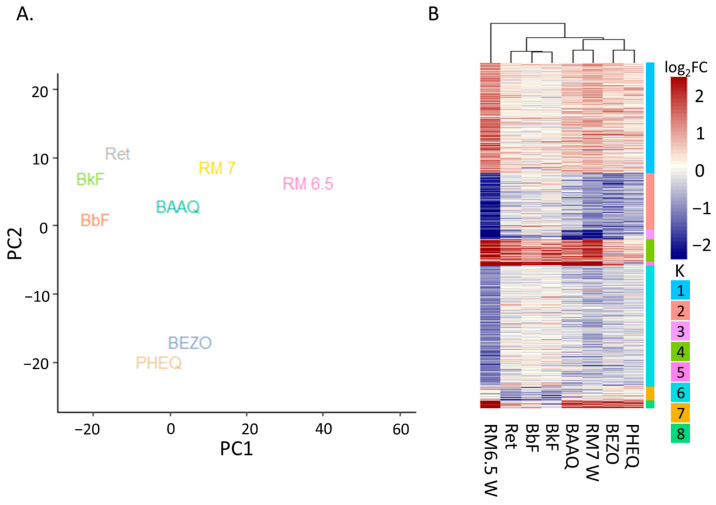
Comparison of the DEGs from the PHSS extract treatments and treatment with PAHs or OPAHs. (**A**) PCA analysis using DEGs of each treatment. PC1 accounts for 53% of variance, PC2 accounts for 20 % of variance. (**B**) A heatmap displaying log_2_FCs for each gene significant in at least one of the treatments (│log_2_FC│ > 1 and p_adj_ <= 0.05). Rows are grouped by k-means (k = 8) and columns are clustered by Euclidian distance.

**Figure 6 toxics-11-00201-f006:**
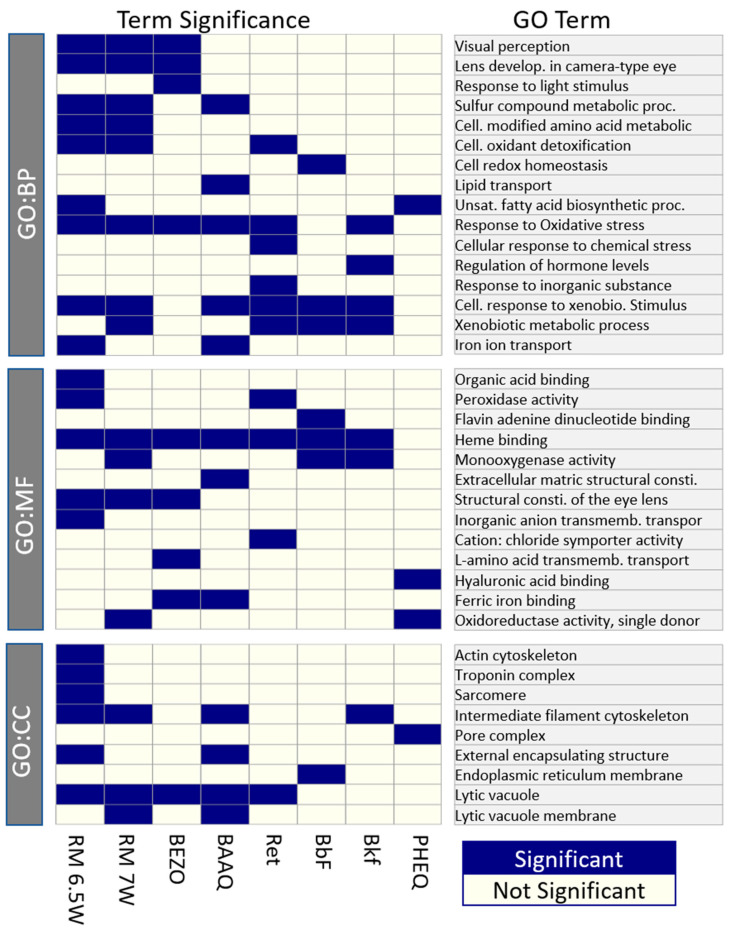
The significant GO terms from the reduced redundancy GO set. Enrichment was tested for the DEGs meeting the criteria of |log_2_FC| > 0.5, p_adj_ < 0.5 for each exposure. Column order was chosen to simplify the comparison between the PHSS extracts and individual chemical exposures. Rows are split by the GO database then clustered by the Jaccard distance of shared genes.

**Table 1 toxics-11-00201-t001:** Results of the 1201 chemical screen for RM 7W and RM 6.5W. Positive identification is indicated by an X.

	Detection	
Chemical	RM 7W	RM 6.5W
o,p’-DDD	X	X
p,p’-DDE	X	X
Hexachlorobenzene	X	X
PCB65	X	
PCB118X	X	
Tonalide	X	
Benzoflourenone	X	
benzanthrone	X	

## Data Availability

The data presented in this study are available in the Appendix A or upon request.

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
