# Peer review of "Coupling Environmental Whole Mixture Toxicity Screening with Unbiased RNA-Seq Reveals Site-Specific Biological Responses in Zebrafish"

_toxics, 2023, doi:10.3390/toxics11030201_

Round 1

Reviewer 1 Report

This study tested an interesting question if the coupling whole mixture screening test and RNAseq could separates and identifies the toxicity and toxicants using superfund site soil samples.

The manuscript was well written and the experiments were performed well; however, there are points to be answered and modified before the manuscript get accepted.

1. Phenotypical responses are stronger in RM 6.5 that contains higher PAHs level. No experiments to prove the suggested main causal toxicants, OPAHs, seems the biggest limitation of this study. Please describe the limitation in discussion.

2. PAHs measured from RM 6.5 had higher concentrations than RM 7.5 and it is matched with the higher toxicity from RM 6.5. On the other hand, the gene expression response implies that the responses are more related to OPAH than PAH. Can you provide potential reasons in the discussion section, why RM 6.5 will have more OPAHs than RM 7.5 although only measured two OPAH were found from RM 7.5?

3. In introduction, genes related to PAH toxicity were introduced but the genes were briefly discussed by describing the differential gene expression heatmap (Fig.5). Please provide more detailed comparison between the samples for the important genes. A graph or numbers in a table for the gene expressions would strengthen the given conclusion.

4. Figure 1B seems better to be drawn with ratio values than the absolute concentrations of PAHs as the descriptions are about the ratio similarity. The current figure could be moved to the supplementary information as it is still informative result.

5. Change the figure 2 A with a point or bar plot showing the distribution of measurements. The current figure only depicts the average and has no error bar or distribution.

6. Heat map on Figure 3c shows very similar pattern of gene expression between the samples. Readers could wonder if the gene expressions known to be related to OPAHs exposure should be reconsidered, considering the similarity between samples matches with the similarity of PAHs ratio. Please cite multiple studies to further discuss if the OPAHs responsive gene expressions are reliable(with consistent results) or questionable.

Author Response

Response to Reviewer 1 comments.

  1. Phenotypical responses are stronger in RM 6.5 that contains higher PAHs level. No experiments to prove the suggested main causal toxicants, OPAHs, seems the biggest limitation of this study. Please describe the limitation in discussion.

See below.

  1. PAHs measured from RM 6.5 had higher concentrations than RM 7.5 and it is matched with the higher toxicity from RM 6.5. On the other hand, the gene expression response implies that the responses are more related to OPAH than PAH. Can you provide potential reasons in the discussion section, why RM 6.5 will have more OPAHs than RM 7.5 although only measured two OPAH were found from RM 7.5?

These were good points. It was not our intention to claim that OPAHS were the causal toxicants, only that the gene expression appeared more like the OPAHs than the PAHs. In response we’ve tried to clarify our position and some possible reasons for the observed gene expression in lines 587 -592. We also added a sentence in the conclusion conceding that it would likely take affects directed analysis and more extensive chemistry to nail down the causal toxicants.

  1. In introduction, genes related to PAH toxicity were introduced but the genes were briefly discussed by describing the differential gene expression heatmap (Fig.5). Please provide more detailed comparison between the samples for the important genes. A graph or numbers in a table for the gene expressions would strengthen the given conclusion.

In generally we tried to point out the important genes as they were relevant in the results / discussion, but the paper intentionally pays more attention to overall expression trends because we find this is more useful for comparing the exposures. The new draft includes supplementary table ** to facilitate easier comparison of these genes for the reader.

  1. Figure 1B seems better to be drawn with ratio values than the absolute concentrations of PAHs as the descriptions are about the ratio similarity. The current figure could be moved to the supplementary information as it is still informative result.

Thank you for the suggestions. Our newest draft revises this figure accordingly.

  1. Change the figure 2 A with a point or bar plot showing the distribution of measurements. The current figure only depicts the average and has no error bar or distribution.

We revised the plots in two ways:

  1. We removed the DMSO control points and replaced them with hashed lines to indicate the levels of response in DMSO control exposures.
  2. We originally hesitated to add error bars because the indicated levels at each point are from 40 fish tested on two separate plates. The new plot includes error bars calculated utilizing a binomial distribution from n = 40 at each point.
  3. Heat map on Figure 3c shows very similar pattern of gene expression between the samples. Readers could wonder if the gene expressions known to be related to OPAHs exposure should be reconsidered, considering the similarity between samples matches with the similarity of PAHs ratio. Please cite multiple studies to further discuss if the OPAHs responsive gene expressions are reliable (with consistent results) or questionable.

Gene expression data for these three compounds is rather sparse in the literature. We also know that OPAHs do not all act the same, so it is hard to use other OPAHs as surrogates. We included a section at lines 580-586 with the two studies we are aware of presenting gene expression in response to the BEZO, BAAQ, or PHEQ. They are consistent with our RNAseq results. We also included a table in the supplemental data.  

Reviewer 2 Report

In this manuscript, Rude et al. performed toxicity assays using developing zebrafish embryos for PSD extracts of two PHSS river mile 6.5W (RM 6.5W) and river mile 7W (RM 7W). They found that RM 6.5W is more toxic, causing defects in notochord development in developing embryos. They further performed transcriptome analysis to identify and compare differentially expressed genes under both treated conditions, as well as to those of individual chemicals.

Overall, the conclusions are well supported by their analyses. However, I do have a couple suggestions that could potentially improve the manuscript.

1.     Texts of line 243-266 were almost duplicated, and should be fixed.

2.     I think the authors performed experiments using 1%, 0.2%, 0.04%, 0.008% of PSD extracts in Figure 2A, but there were 5 data points shown in Figure 2A. I’m confused by the least concentration they used, and it was not clear to me what this is.

3.     In the same Figure 2A, it would be better to add results of DMSO treatment to show no major effect for mortality and phenotypes from DMSO group. In addition, if the phenotypic scoring/counting were performed multiple times as replicates, error bars should be added to the plot.

4.     In Figure 2B and texts, the authors showed strong defects in notochord development, yet no images were provided to demonstrate these defects. I think the manuscript could be strengthened by providing some images of developing embryos, particularly to show notochord developmental defects.

5.     In Figure 3A, the authors showed overlap between differentially expressed genes of RM6.5W and RM7W. Though it became more clear with panel B and C that roughly equal increased and decreased gene expression. It might be worthy to present venn diagrams of both differentially increased and differentially decreased genes.

6.     Text in Line 610 was not complete, and needs to be fixed.

Author Response

Response to reviewer 2 comments.

  1. Texts of line 243-266 were almost duplicated, and should be fixed.

I’m sorry we did not catch this. Thank you for pointing it out. We’ve removed the repeated section in the newest manuscript.

  1. I think the authors performed experiments using 1%, 0.2%, 0.04%, 0.008% of PSD extracts in Figure 2A, but there were 5 data points shown in Figure 2A. I’m confused by the least concentration they used, and it was not clear to me what this is.

The point closest to the y-axis in both plots of 2A is the DMSO control. You’re right that this is misleading though because the DMSO controls are technically un-plottable on a log scale. Apparently ggplot just choose a point near or con the y axis to plot a log(0). The response to your third comment provides more detail. 

  1. In the same Figure 2A, it would be better to add results of DMSO treatment to show no major effect for mortality and phenotypes from DMSO group. In addition, if the phenotypic scoring/counting were performed multiple times as replicates, error bars should be added to the plot.

We revised the plots in two ways:

  1. We removed the DMSO control points and replaced them with hashed lines to indicate the levels of response in DMSO control exposures. We also revised the short exposure summary sentence in the results section to make this more clear.
  2. We hesitated to add error bars because the indicated levels at each point are from 40 fish tested on two separate plates. The new plot includes error bars calculated utilizing a binomial distribution from n = 40 at each point.

  1. In Figure 2B and texts, the authors showed strong defects in notochord development, yet no images were provided to demonstrate these defects. I think the manuscript could be strengthened by providing some images of developing embryos, particularly to show notochord developmental defects.

Agreed. Unfortunately, the dev tox assays were done years ago and none of the images were retained. There are images from a paper our group previously published that are from the same mixture exposures. The newest draft references these so the reader can find them if they are curios.  

  1. In Figure 3A, the authors showed overlap between differentially expressed genes of RM6.5W and RM7W. Though it became more clear with panel B and C that roughly equal increased and decreased gene expression. It might be worthy to present venn diagrams of both differentially increased and differentially decreased genes.

We previously considered breaking the Venn Diagram into two as you described, but decided the extra space it would require was not warranted because the heatmap and scatter plot indicate the relative proportions of genes with increased and decreased expression. Respectfully, we would prefer to keep this figure as is.

  1. Text in Line 610 was not complete, and needs to be fixed.

Thank you. Our newest draft resolves this.